# Epigenetic Regulation of Circadian Rhythm and Its Possible Role in Diabetes Mellitus

**DOI:** 10.3390/ijms21083005

**Published:** 2020-04-24

**Authors:** Michael Hudec, Pavlina Dankova, Roman Solc, Nardjas Bettazova, Marie Cerna

**Affiliations:** 1Department of Medical Genetics, Third Faculty of Medicine, Charles University; Ruská 87, 100 00 Prague, Czech Republic; nardjas.bettaz@gmail.com (N.B.); marie.cerna@lf3.cuni.cz (M.C.); 2Department of Anthropology and Human Genetics, Faculty of Science, Charles University; Viničná 7, 128 00 Prague, Czech Republic; pavlina.dankova@natur.cuni.cz (P.D.); solc.roman@seznam.cz (R.S.)

**Keywords:** type 2 diabetes mellitus, circadian clock, epigenetic regulation, metabolic syndrome, sleep, aging

## Abstract

This review aims to summarize the knowledge about the relationship between circadian rhythms and their influence on the development of type 2 diabetes mellitus (T2DM) and metabolic syndrome. Circadian rhythms are controlled by internal molecular feedback loops that synchronize the organism with the external environment. These loops are affected by genetic and epigenetic factors. Genetic factors include polymorphisms and mutations of circadian genes. The expression of circadian genes is regulated by epigenetic mechanisms that change from prenatal development to old age. Epigenetic modifications are influenced by the external environment. Most of these modifications are affected by our own life style. Irregular circadian rhythm and low quality of sleep have been shown to increase the risk of developing T2DM and other metabolic disorders. Here, we attempt to provide a wide description of mutual relationships between epigenetic regulation, circadian rhythm, aging process and highlight new evidences that show possible therapeutic advance in the field of chrono-medicine which will be more important in the upcoming years.

## 1. Introduction

Almost every cell in the organism carries its genetic information, containing thousands of genes. As an individual develops and grows, their cells divide and differentiate. Tissues and organs are gradually being synthesized, and in each organ, the cells specialize to perform only one function. For each cell, the initial deposit of several thousand genes is enough. They are never used all at the same time [1,2]. There are genes that play an important role in regulating circadian rhythms [3]. They are referred to as clock-controlled genes (CCGs). CCGs are necessary genes for cell survival, actually for the whole organism [4]. Perhaps the most important genes, their products respectively, include: circadian locomotor output cycles kaput (CLOCK) and its paralog neuronal protein with PAS domain 2 (NPAS2), brain and muscle aryl hydrocarbon receptor nuclear-translocator like Arntl (BMAL), period (PER), cryptochrome (CRY), retinoic acid-related orphan receptor (ROR), sirtuin (SIRT), nuclear subfamily 1 receptor group D member 1 (NR1D1 or REV-ERBα), lysine specific demethylase 1A (KDM1A), histone deacetylases (HDACs), ubiquitin ligase: F-box and leucine-rich repeat protein 3 (FBXL3), nuclear receptor corepressor 2 (NCOR2) and nicotinamide phosphoribosyltransferase (NAMPT). These are involved in several regulatory loops designed to maintain the stability of the organism [3,5]. These loops are shown in Figure 1. The circadian regulatory pathways are complex and include robust molecular negative and positive feedback loops of transcription factors, genes and epigenetic modulators. There is a central circadian clock in the CNS which is connected with peripheral circadian clocks in the peripheral tissues by the endocrine system. It is worth mentioning that the endocrine system plays a major role in regulating physiological processes and, importantly, is controlled by circadian molecular mechanisms [6].

Processes such as sleep, locomotion, nutrient uptake and metabolism of substances, immune responses, excretion, regeneration, and more must take place with circadian rhythmicity. This is expressed as Chronotype, the time of day preference for daily activities. There are several chronotypes. Extreme preferences are morningness and eveningness. The individual preference of daily cycle is set up by a combination of genetic and epigenetic factors and has influence on physical and mental health [8,9]. For example, cortisol levels differ in two chronotypes (early and late). Early chronotype is associated with a more pronounced adreno-cortical activation after awakening and conversely, less pronounced adreno-cortical activation is a characteristic of the late chronotype. [10]. Due to disruption of cortisol rhythms patients often suffer from sleep and circadian rhythm disorders [6,11]. Another example is management of melatonin. Melatonin secretion is controlled by the SCN pacemaker and is important for synchronizing peripheral tissue clocks [12]. This means that every single cell must be fine-tuned to carry out its functions at an appropriate time. These processes can also be run in another part of the day, but at that point, they are executed under suboptimal conditions (for example, impaired glucose tolerance or low subjective alertness) [13,14]. That’s why CCG are so important for the proper functioning of the organism. These genes set for all individuals their inner clock. Thanks to the CCG, cells of the body are constantly synchronized and consequently‚ according to their settings, ready to perform their function in coordination with other cells. The connection between inner molecular clock and physiological output is shown in Figure 1. Impaired synchronization and functional coordination among body cells may result in a higher susceptibility to both acute and chronic diseases such as type 2 diabetes mellitus (T2DM) and metabolic syndrome [14,15,16,17,18]. These disorders are more common in the elderly. Aging leads to less robust circadian rhythms and less robust circadian rhythms lead to accelerated biological aging which predisposes to chronic diseases [19]. CCGs adjust the internal environment so that the organism can function optimally. Hierarchically, there are mechanisms that synchronize CCGs with the outer environment (for example timing of food intake) [20,21,22].

An insufficient concentration of CCG proteins or a delay in reaching the concentration threshold for activation of other genes contributes to the disruption of regular oscillation in peripheral tissue activity. It has been found that virtually all major genes involved in the regulation of circadian loops, in their deficiency, induce a change in the circadian period and have an impact on health condition (Table 1). For example, an abnormally elevated level of PER protein shortens the daily period [23,24]. Any change in the circadian period may affect the quality of sleep. The quality and length of sleep are demonstrably related to many disorders, which are due to worse regeneration and an insufficient body rest [14,18,25].

## 2. Genetics of Type 2 Diabetes Mellitus

Type 2 diabetes mellitus (T2DM) develops as a result of progressive disorder in both mechanisms, insulin secretion and insulin resistance [37]. The loss of insulin secretion occurs by a mechanism other than autoimmunity, and the process is unlikely to result in complete loss of β cells. One such important mechanism is the apoptosis of β cells due to metabolic (high levels of lipids and glucose) and nervous (stress) disbalance. The presence of both aberrations is a prerequisite for clinical manifestation. Genetic predisposition and a number of external defined factors such as obesity, stress, low physical activity and smoking contribute to the disease. T2DM is a multifactorial disease with a complex inheritance where multiple genetic loci and environmental factors are implicated [38,39]. Hereditary genetic factors can be either predisposing to the disease or protective against it. The risk of T2DM manifestation is high and is dependent on the population. This risk is reported to be in the range of 1–5% in most populations. It is highest among ethnic groups of American blacks, Mexicans, and Indians (the Pima in Arizona), the lowest among Eskimos [37,40,41]. It is low in the Scandinavian countries, yet relatively high in southern Europe. T2DM is most often manifested in men between 45 and 65 years and in women between 50 and 55 years. The average risk of manifestation is 10% for siblings and 20% for children of a diabetic proband. The probability of concordance in monozygotic twins is between 60–90%, in dizygotic 24–40% [42]. Concordances in monozygotic and dizygotic twins are significantly higher compared to T1DM. In addition to obesity, other components of the metabolic syndrome often interfere with the phenotypic manifestation of the disease. The identification of predisposing genes is hampered by the presence of a large number of internal and external environmental factors that affect insulin sensitivity and secretion, such as age, gender, ethnicity, physical fitness, diet, smoking, obesity, fat distribution, pregnancy, stress and last but not least sleeping disorders [43,44]. Dysregulation of melatonin, the hormone controlling circadian rhythm that is produced from the epiphysis, increases insulin resistance [45].

Chronic diabetic complications are divided into macrovascular (atherosclerosis) and microvascular (diabetic nephropathy, diabetic retinopathy, and diabetic neuropathy). The most common complication of T2DM is diabetic nephropathy (DN). It is also the major cause of chronic renal insufficiency in the United States and Western Europe and it is a major cause of morbidity and premature mortality in patients with type 1 and type 2 diabetes. The prevalence of DN ranges from 25% to 40% after 25 years of diabetes in patients with T2DM. This variability is due to the fact that T2DM may remain undiagnosed for a long time. The basic prerequisite for the development of DN is the long-term poor compensation of the underlying disease. The proportion of genetic predisposition to DN is evident from a number of epidemiological studies, where the cumulative risk for developing DN in relatives of nephropathy probands is 50% higher than in ones of diabetic probands without nephropathy [40,41,46,47,48,49,50].

## 3. Epigenetics of Type 2 Diabetes Mellitus

The different types of epigenetic modifications are closely connected to the regulation of various cellular processes and often amplify themselves. The hallmark of epigenetic variability, compared to genetic variability, tends to be acquired in a gradual rather than a sudden process. Changes in epigenetic status are particularly characteristic for the aging process [51]. These changes occur randomly in the tissue and initiate a high degree of epigenetic cell variability. Epigenetic mechanisms are essential throughout the life of the cell and organism for the integration of environmental signals, both endogenous (autocrine, paracrine and endocrine) or exogenous from the external environment [52]. In particular, early embryonic development is the most important period of establishment and maintenance of epigenetic markers [53]. In this respect, the results of a Dutch study are very important, pointing out the importance of prenatal exposure to certain environmental factors that can set the profile of epigenetic modifications, thus affecting metabolic pathways and the entire physiology of the organism and affecting predisposition to chronic adult diseases [54]. The Dutch study is in agreement with the theory of thrifty phenotype, which is set by epigenetic mechanisms at a time of starving [55]. A significant association between different levels of methylation of DNA genes involved in metabolic and cardiovascular diseases and growth signaling has been found. For example, the INS-IGF locus includes the insulin genes (INS) and the insulin-like growth factor 2 (IGF2) where reduced methylation was detected. Conversely, increased methylation was found, for example, for the interleukin 10 (IL-10), leptin and GNA S antisense RNA genes. Insufficient methylation potential of the cell has led to insulin resistance and hypertension in the offspring in a sex-dependent manner [56]. Higher levels of DNA methylation were found in men rather than women [57]. Interactions between sex hormones and expression of DNA methyltransferases have been observed, which could explain gender differences. Progesterone and estrogen reduced the expression of DNA methyltransferases in the endometrium [58]. Studies showing that genetically identical twins differ significantly in terms of DNA methylation and histone modifications also yield significant results [59]. Only 65% of monozygotic twin pairs show almost identical 5-methylcytosine content in genomic DNA and identical histone acetylation profiles of H3 and H4. The differences correlate with age and time spent together. Young twin couples are more similar than older couples, and those who spend more time together are more similar than those who have lived separately. This study clearly demonstrates that the epigenotype of genetically identical individuals varies dramatically and that the extent of divergence is modified by the environment and increases with time. This would explain the discordance of diseases in monozygotic twins, the variability of their onset time and their severity [60]. Thus, RNA and protein expression of genes can be altered by epigenetic changes that are induced by the environment. The main environmental factors that affect the organism are cyclic changes that are responsible for the creation of circadian rhythms. Circadian rhythms arise from 24-hour day-night light-dark cycle that is caused by the rotation of the Earth. Accordingly, circadian rhythms accompany the evolution of most forms of life. Almost all organisms adapt the geophysical cycle to their biological cycles: 1) sleep - wake, 2) hormone secretion, 3) body temperature and blood pressure, 4) motor activity, and 5) fasting - feeding.

## 4. Epigenetic Control of Circadian Rhythms

Light is the strongest stimulus for circadian rhythms. When the sun rises in the morning, we receive this signal through the eye. The inner layers of the retina contain ganglion cells, the intrinsically photosensitive retinal ganglion cells (ipRGC) [61,62]. Under the influence of long-term light signal uptake, the so-called ipRGC maintain a depolarized state and thus generate an action potential signal that arrives at the ventromedial group of neurons in the suprachiasmatic nucleus (SCN) [63]. Here, CCGs expression is activated and the overall organism synchronization with the external environment is activated. The main molecular loops are discussed in order below.

### 4.1. Regulatory Genes CLOCK and BMAL1

The clock genes *CLOCK* (*NPAS2*) and *BMAL1* are probably the most important regulators of circadian rhythms. After their translation and activation, they trigger the transcription of other CCGs. The main feedback loop in which CLOCK and BMAL1 [64,65] operate includes two other important controllers: PER and CRY [21,66]. After transcription of genes for the CLOCK and BMAL1, both transcripts are transported to the cytosol where they are translated. CLOCK after translation acetylates BMAL1 on Lys 537 and then both dimerize together [67,68]. Dimerization is enabled by the helix-loop-helix motif on the per-arnt-sim domain (PAS domain) of the CLOCK protein. Further epigenetic alterations, sumoylation, and phosphorylation of BMAL1 [68,69] enhance the functionality of both proteins. After dimerization, they enter the nucleus and bind to the promoter region of the promoter enhancer cassette (E-box), which is common to all CCGs, and trigger the expression of the *PER* and *CRY* genes [65]. The *CRY* and *PER* genes transcripts are then translated in the cytoplasm and, upon reaching the threshold level, are post-translationally phosphorylated and transported to the nucleus, where they form a complex upon entry [24]. They then inhibit their own transcription, activated by CLOCK and BMAL1 regulators [70]. Thereafter both PER and CRY are degraded in the proteasome. DNA methylation has important role in regulating *PER2* gene and dysregulation leads to disruption of the metabolism [71]. PER is believed to be phosphorylated by casein kinase I (CKI) and CRY by adenosine monophosphate activated protein kinase (AMPK) [72]. The mutation in CKI leads to a shortening of the daily period because the undegraded PER protein accumulates and its high levels ultimately result in a circadian rhythm acceleration [73]. Degradation is controlled by ubiquitination. FBXL3 thus modifies CRY post-translationally [74]. A gene-based approach detected a significant association of CpG methylation pattern in *PER2* gene with both blood glucose and insulin resistance [75] There is a similar observation for *CLOCK* and *BMAL* genes. Another result suggests that Bmal1 controls gene expression in response to inflammatory activation by regulating the epigenetic status of enhancers [76]. There is an evidence that particulate air pollution exposure during gestational life change methylation status of core circadian factors (CLOCK-BMAL1) and may be connected with circadian disruption [77].

### 4.2. Regulation of BMAL1 Expression

*BMAL1* gene expression is itself controlled by its own regulatory loop. This is just a mechanism supporting the stability and robustness of the entire system. Two genes are involved in this pathway: *ROR* and *REV-ERB* [29]. Both belong to the CCG group and their expression is also activated by binding of CLOCK-BMAL1 to the E-box promoter region of these genes. Both are also translated in the cytoplasm after their transcription and then transported back to the nucleus. In the nucleus, ROR and REV-ERB compete for ROR responsive element (RRE), the *BMAL1* gene promoter region [78]. While ROR activates transcription, REV-ERB serves as an inhibitor but also blocks expressed CRY [78]. The entire inhibitory complex is stabilized by binding of NCOR2 through the heme molecule of REV-ERB [79]. Subsequently, HDAC3 is brought to RRE site to cause chromatin condensation [80].

### 4.3. The Role of SIRT1 in Regulating Circadian Rhythms

SIRT1 is a NAD^+^-dependent protein deacetylase. In aging individuals, the SIRT1 level in the SCN decreases. Its main role in the regulation of CCG is not yet clarified. SIRT1 is known to help activate BMAL1 transcription [81]. On the other hand, SIRT1 deacetylates BMAL1 at Lys-537 and thereby disrupts the CLOCK/BMAL1 complex [82]. Furthermore, it was observed that SIRT1 has the capacity to deacetylate PER1, which will further result in its degradation in the proteasome [83]. However, SIRT1 is irreplaceable for the functioning of the control loops. For example, BSKO mice strain (Sirt1 knockout strain) exhibit the same circadian period disruption as that seen in elderly individuals [84]. Because SIRT1 is NAD^+^-dependent, it is directly reliant on its concentration. NAD^+^ is supplemented with NAMPT, which belongs to CCG. The highest expression level of NAMPT (in mice) is at the end of the day [5].

## 5. Circadian Rhythms and Aging

Aging of an organism can be defined as a condition where repair mechanisms are no longer able to repair all degenerative changes. As mentioned, much of the DNA sequence is in the condensed state most of the time. This arrangement probably protects the DNA from external and internal factors that could damage it. However, during expression, some of the DNA is decondensed and is more susceptible to damage [85]. Moreover, during one stem cell division, there are statistically 7.6 × 10^−10^ mutations [86]. DNA can also be damaged by external physical and chemical factors, mutagens. It is precisely the state of condensation that regulates epigenetic mechanisms such as acetylation, methylation, and opposite processes, both at the level of histones and DNA itself.

Internal damage factors include metabolic products, for example reactive oxygen species (ROS), which are produced in mitochondria and peroxisomes. ROS can cause breaks and other DNA damage. This condition is called oxidative stress. An example can be the oxidative damage to the guanine nucleotide, which is converted to 8-oxoguanine under the influence of oxidative stress [87]. 8-oxoguanine accumulates in the DNA sequence with age and this accumulation becomes severe around the age of 40 years. The concentration of 8-oxoguanine in promoters of genes with important neural function is already so high that their expression decreases [87,88].

A significant group of enzymes interfering with the chromatin structure are the aforementioned sirtuins. So far, seven homologs are known in mammals. In circadian rhythms, the most important homolog is Sirt1. All sirtuins are NAD^+^-dependent and their activity is regulated by the current NAD^+^ concentration. Mammalian Sirt1 is most closely related to Sir2, found in the fruit fly. Higher Sir2 expression has been shown to lead to an increase in fruit fly’s life span [89], while mammals with knock-out Sirt1 homolog die very early due to developmental defects [90]. Thus, it is clear that chromatin and sirtuin modifications play an important role in genome stability. Today Sirt1 is perceived as a basic epigenetic regulator that protects the organism from the influences that lead to aging. The decrease in NAMPT activity during aging also reduces gene deacetylation. These are then more accessible to transcription factors and generally more susceptible to damage due to the absence of protein scaffold protection. Worse DNA protection leads to a gradual loss of the genome integrity and irreversible DNA damage. This may be manifested by premature aging. An example of insufficient DNA protection in aging is premature aging syndrome which is caused by non-functional LAMIN A and dysfunctional helicase that has the task of loosening the DNA structure similar to the situation of loosening histones during transcription [91]. The result is an unstable genome with many mutations reminiscent of a few decades (years).

There is an evidence that long term shift work exposure may lead to a range of modifications in methylation of DNA and aging acceleration. There is an association between long term shift work and changes in DNA methylation of *ZFHX3* gene that encodes a protein acting in circadian rhythm pathways [92].Finally, important evidences support a bidirectional model of mutual relationship between aging processes and molecular circadian mechanisms [93]. Deletions in *Bmal1* and *Period* genes seem to accelerate aging in *Drosophila* and mice models [94,95] and fetal transplants that contain the SCN can restore circadian rhythms and increase lifespan in older animals [96,97].

## 6. Interindividual Variability of Circadian Preference

Most people prefer the daily rhythm when they get up between 6 am and 8 am. Naturally then, they go to bed between 10 pm and 12 pm [98,99]. Extremes of this distribution are people who have the so-called Advanced Sleep-Wake Phase Disorder or Delayed Sleep-Wake Phase Disorder. It is important to note that the natural preference for a certain circadian period does not necessarily lead to health problems. The problem arises when particular individuals are forced by their surroundings (the working environment) to disregard a circadian period that is natural to them. But natural preference of eveningness leads to dysregulation despite following preferred rhythm [100]. It was found that each hour delay in mid-sleep time on free days (a metric of chronotype), was associated with significantly higher HbA1c. Later chronotype and larger dinner are associated with poorer glycemic control [101].

Further, it was observed that delayed sleep phase leads to a higher prevalence of type 2 diabetes, higher risk of arterial hypertension and lower fasting serum levels of total and LDL cholesterol [100,102,103].

Thus, the risk factor itself is the lifestyle in the suboptimal circadian period. In addition to people with an extreme circadian period, it also affects people with a normal circadian period if their working hours shift to that part of the day when their organism is already naturally preparing for the sleep phase. Variability in sleep timing is more important than variability in sleep duration [104,105]. There are night-time jobs and jobs requiring frequent traveling across several time zones. In this case, the organism is artificially exposed to another circadian period. Activities are shifted to the evening due to modern lifestyle. This generates a discrepancy between the preferred sleep timing and the social sleep opportunity [105]. The gradual disruption of the organism due to disrespect of the natural circadian period is treacherous because its consequences will only be manifested after several decades [105]. This probably could be the reason for the growth of the so-called civilization diseases over the last 50 years. The organism initially seeks to cope with a suboptimal circadian period. However, after a longer period of time, compensation is no longer sufficient and the first signals of imbalance begin to emerge. These include poor sleep, inability to maintain long-term alertness/long-term fatigue or frequent morbidity and increased risk for the development of T2DM [14,18,25].

### 6.1. Advanced Sleep-Wake Phase Disorder

The Advanced Sleep-Wake Phase Disorder (ASWPD) manifests as a change in the time when natural daily rhythms take place. Under certain circumstances, people with ASWPD preference do not suffer from reduced sleep time or overall poor quality. They go to bed naturally around 7 pm and spontaneously wake up around 4 am. This shift to the natural preference of another circadian period is probably due to the interaction of PER and CKI [23,32]. After exposure to light during the night, only the PER1 and PER2 homologs react appreciably in the SCN, and these two also differ in their reactions. While PER1 reacts acutely and its RNA level quickly returns to normal, PER2 reacts insidiously to achieve the same level of RNA, if not higher. These properties make them the major candidates for the ASWPD phenotype [20]. So far, only the disorder related to PER2 interaction with CKI is known. In its natural state, CKI regulates PER2 by phosphorylating it post-translationally. This post-translational mark is the key for subsequent PER2 degradation. Phosphorylation cannot occur if the sequence recognized by CKI at PER2 is mutated or when a mutation occurs in CKI itself [73]. It is then unable to phosphorylate PER2 [23,32]. If these two proteins do not interact, PER2 may accumulate and accelerate the circadian period. The result may be an ASWPD-like phenotype. Under certain circumstances, these people may have health problems due to different daily activity preferences [15].

### 6.2. Delayed Sleep-Wake Phase Disorder

The Delayed Sleep-Wake Phase Disorder (DSWPD) is a phenotype with the opposite manifestation than ASWPD. It is also a shift in the natural circadian rhythm, but here the preference for the beginning of natural daily rhythms is shifted to the late morning hours. People exhibiting this phenotype go to sleep after 2 am and sleep until 10 am [33,106]. The length and quality of sleep do not change significantly either. As with ASWPD, there is a risk of health problems occurring under certain circumstances. This phenotype is probably caused by the PER3 gene. There are four PER3 haplotypes varying in different amount of variable number of tandem repeats (VNTR). DSWPD is most commonly found in the H4 haplotype [107]. This haplotype contains only four repeats. In addition, the V647G substitution mutation occurs in this shorter haplotype. A similar S662G mutation affects the ability of CKI to phosphorylate PER2. It is likely that PER homologs carry a binding site around the amino acid at position 650 that is recognized by CKI and which is important for the ability of CKI to phosphorylate PER [23]. It is possible that this polymorphism also plays a role in PER3 phosphorylation by CKI. Interestingly, the PER3 haplotype containing five repeats occurs more frequently in patients with the opposite, i.e., morning preferences [108]. Gene-based approach detected a significant association of CpG methylation pattern in *PER2* gene with both blood glucose and insulin resistance [75] There is a similar observation for *CLOCK* and *BMAL* genes. Delayed Sleep-Wake Phase Disorder phenotype is associated with a higher prevalence of type 2 diabetes, higher risk of arterial hypertension and lower fasting serum levels of total and LDL cholesterol [102] and poorer glycemic control [101]. A relationship was found between epigenetic changes and nightwork. Differences were detected in the gene body of *PER3* [109]. Another study reveals a positive association between an average sleep duration of less than 6 h and the methylation pattern of *PER2* and *CRY2* [110].

## 7. Pathology Resulting from Decompensation of Circadian Rhythms

### 7.1. Metabolic Syndrome, T2DM and Circadian Rhythm Disorders

Metabolic syndrome is a disease associated with modern lifestyle. Its signs and symptoms can be seen in both, people living in developed and developing countries [15,17,111]. Fast lifestyle, long working hours and working hours in the evening, irregular eating habits or evening overeating, sleep deprivation, insufficient exposure to natural daylight and, on the contrary, frequent exposure to artificial light are all possible causes of dysregulation of circadian rhythm and circadian clock desynchronization between the central nervous system and the periphery. The signs and symptoms of this syndrome are obesity, hyperglycemia, hypertension and elevated triglycerides [111]. Diseases, including diabetes, psychological, neurological, respiratory and gastrointestinal/abdominal disorders are connected with eveningness [100]. Unhealthy diet, lower levels of physical activity, longer sedentary time, and poor sleep habits are also connected with eveningness [103]. In addition, there may be a causal relation with the higher incidence of autoimmune diseases. According to a research summary published in the USA, there is an association between the average shorter sleep time and the increasing incidence of obesity and diabetes over the same period of time summarized in [15]. Other research suggests that poor quality of sleep may probably be related to the development of T2DM with typical signs and symptoms [14,18].

Likewise, circadian period is synchronised by timing of food intake [112]. There is an evidence that food timing is inherited. The genetic component has a greater effect on timing of breakfast. Conversely, the environment is more responsible for timing of lunch or dinner [113]. Excessive adipose tissue in obese individuals has additional health effects. Fat tissue produces and secrets hormones that are collectively called adipokines. They have the task of maintaining balance in the body’s energy system, but they are also involved in the regulation of the immune system and sleep. Leptin (a satiety-inducing hormone) was discovered as the first adipokine [114]. Its daytime fluctuation is impaired in people suffering from obesity [115]. Low leptin levels are also associated with poorer sleep quality [116]. Indeed, long-term sleep deprivation reduces leptin levels and increases ghrelin (gastric hormone) levels [25]. People suffering from sleep apnea tend to develop metabolic syndrome and, conversely, treatment of this disease leads to metabolic balance and health improvement [17,117].

Another factor naturally involved in regulating circadian rhythm is melatonin. Melatonin seems to have its place in regulating glucose levels. Melatonin has two receptors: MTNR1a and MTNR1b [118]. MTNR1b is expressed in the SCN [119], but Langerhans islets and pancreatic β-cells [120] have been identified as additional sites of expression. Melatonin blocks insulin secretion, which is normally stimulated by glucose [121,122]. When melatonin synthesis is triggered and its level is rising in the body, natural glucose processing after its uptake is disrupted.

Further, SIRT1 indirectly affects the metabolism of fats and sugars because it is associated with the regulation of adipocyte differentiation, gluconeogenesis and insulin secretion [34,35,36]. Furthermore, it promotes fat burning during periods of lower food intake (sleep) due to binding to the peroxisomal activated receptor (PPARγ) [34].

CLOCK and BMAL1 play a role in protecting the body against symptoms of metabolic syndrome. The CLOCK gene mutation leads to the development of hyperphagia, hyperlipidemia, hyperinsulinemic hyperglycemia and sleep disorders [26]. During the expression of BMAL1 in fibroblast cell line 1 with a phenotype similar to the adipocytes (3T3-L1), there is an induction of lipogenesis and, conversely, the inhibition of BMAL1 in the same cell line leads to adipogenesis [27].

### 7.2. The Role of REV-ERB Nuclear Receptors in Metabolism Regulation

The nuclear receptor regulatory loop has already been introduced. Importantly, the complex of three proteins (REV-ERBα, NCOR, HDAC3) inhibits the transcription of the BMAL1 gene at night. Deletion of the HDAC3 gene causes disruption in lipid liver homeostasis and severe steatosis, while deletion of the REV-ERBβ gene does not cause significant disruption. However, a higher level of BMAL1 is evident [31,80,123]. The REV-ERBα homolog is probably dominant in proper function. Its deletion results in a shift in circadian rhythm. By excluding both REV-ERB homologs, regulation of BMAL1 and NPAS2 expression is fundamentally affected. This suggests that both of these homologs are promoted to function properly in the REV-ERB-NCOR-HDAC3 complex. It is possible that this system works as a backup. Thus, if the function of one of the homologs is violated, the other is able to perform this function [29]. Interestingly, there are more genes involved in regulatory loops of circadian rhythms that have these backups in the form of homologs [29,124].

### 7.3. Circadian Rhythm of Blood Pressure

Maintaining circadian rhythms in the periphery is as important as in the CNS. For example, 10% of the liver genes and 8% of the heart genes exhibit circadian rhythmicity [125]. In an experiment where a group of adults participated in a 6-day test in which they were artificially shortened to 4 h of sleep each day, and then 7 days in which they were artificially extended to 12 h of sleep each day, rhythm disruption resulted in higher glucose and insulin levels, moderate arterial blood pressure, lower leptin levels and reduced sleep quality [15]. The variability in sleep patterns (particularly irregular sleep-onset timing and duration across days) appears to contribute to poor cardiometabolic health [104]. Worse quality of sleep increases the risk of hypertension development [126,127]. This has a long-term adverse effect on the entire cardiovascular system. Both hypertension and hypotension may be due to dysregulation of CCG. Hypotensive conditions can be induced by inhibition of BMAL1 [28]. Hypertensive conditions can be induced by inhibition of CRY under special circumstances [30].

## 8. Conclusions and Possible Treatments

The epigenetic processes regulating circadian rhythms are difficult to modify by using drugs. A problem that will remain unresolved for a long time is that approaches using DNA methylation inhibitors and histone deacetylase inhibitors are non-specific [128,129]. Furthermore, these approaches are directed to the treatment of tumors and therefore are not very suitable for the treatment of circadian rhythms dysregulation. In addition, circadian rhythm disorders are often the result of the intersection of several factors [3]. For these reasons inducing chemical modifications using drugs is unrealizable for near future.

In theory, however, it would be possible to diagnose the circadian rhythm of a particular person. This examination would be similar to the currently used genetic testing for congenital genetic diseases or typing of intrinsic risk alleles of major histocompatibility complex class II (MHC class II) for determination of autoimmune diseases predisposition. Criminalistics now speculate about the use of knowledge of methylation changes in CCG. There are methylation models of important circadian genes that determine at what time their expression peaks approximately. In determining the time of the death of a person, a methylation profile of selected CCGs would be found, which would correspond to the activity time of these CCGs [16,130].

If the patient’s circadian rhythm is determined, this knowledge can be used similarly. However, it is necessary to first obtain the normative graphs and tables of the circadian period of the population with which the patient would be compared. Some data already exist today [98,99]. To be as accurate as possible, a circadian period of healthy people with a preference for daily rhythms without shifts to early morning or late evening hours would be necessary. From this data, benchmarks would be set for the time at which patients would be examined. The examination itself could take place in two ways: by determining the methylation pattern, or by the expression pattern of the selected CCGs. From these results, the shift in the patient’s circadian rhythm versus norm would be evident. The eating and sleeping times should be aligned with patient’s chronotype. This knowledge and practices could be used to treat obesity and T2DM. There is evidence that dysregulations of molecular clocks are connected with the emergence of autoimmune diseases. In these cases, simply balancing the circadian rhythm cannot solve the consequences of those diseases [74,131,132].

Two large randomized studies in obese adults (mean BMI 31) with impaired glucose tolerance achieved a 58% reduction in the incidence of diabetes by lifestyle modification (healthy diet, weight loss and exercise). This result was directly related to lifestyle changes and weight loss [133,134]. However, despite achieving consistent lifestyle modification, approximately 10% of obese adults with impaired glucose tolerance developed T2DM within three years of follow-up. This finding could indicate the presence of inadaptable genetic risk factors or indicate the need for earlier lifestyle intervention [135]. Another study used a new methodology similar to GWAS, named the Environment-Wide Association Study (EWAS) to model the effects of environmental factors likely involved in the etiology of T2DM (pesticides, heavy metals, dietary supplement consumption). A significant risk was found for systemic levels of pesticides derived from heptachlor epoxide, polychlorinated biphenyls and γ-tocopherol (vitamin E), while a protective effect was found for β-carotene (a precursor of vitamin A) [43]. The Diabetes Prevention Program (DPP) has provided supportive evidence that external factors modulate phenotypic expression in high-risk genotypes [134]. This suggests the possibility that lifestyle modification can mitigate the risk created by the genetic background [136]. Another analysis included the genotyping of DPP study participants at up to 34 risk loci for T2DM. The aim was to determine if preventive interventions maintained their efficacy in individuals with a higher genetic risk. Importantly, both metformin treatment and lifestyle modification have effectively reduced the risk of developing diabetes. Interestingly, the lifestyle modification could be even more effective in individuals with the highest genetic risk scores [44].

In conclusion, there are evidences that changes in circadian rhythm, including sleeping pattern, lead to higher risk of developing T2DM. Poor quality of sleep is associated with low leptin levels [116]; obesity, hyperglycemia, hypertension, elevated triglycerides [111] and development of T2DM [14,18]. Importantly, variability in sleep timing indicates to have major effect on sleep and health quality [104]. Changes in DNA methylation in core clock genes CLOCK, BMAL and PER are associated with high blood glucose level and insulin resistance [75]. It was found that long term nightwork is associated with changes in DNA methylation of PER3 gene body [109]. These observation shows connection between circadian (sleep) dysregulations and epigenetics changes that are involved in the development of T2DM. Notably, epigenetic modifications can be inherited [137]. Factors like particulate air pollution [77], preeclampsia [138] or maternal circadian disruption [139] influence DNA methylation pattern in new-born and may have an impact on its health condition. Nonetheless, advances in the field of circadian disruption treatment are promising. Timing of lunch or dinner is more influenced by environment [113]. The therapeutic interventions may be more effective by modifying factors that take place later on in the day. Facer-Childs and colleagues show the ability of a simple non-pharmacological intervention reducing negative elements of mental health and disturbances of sleep by a phase advance of around 2 h. There were significant reductions in subjective ratings of depression and stress. In addition, elements of cognitive (reaction time) and physical (grip strength) performance were significantly improved during ‘non optimal’ times [8]. There is another new approach to chrono-medicine [105]. Roenneberg and Merrow offer a plan how to implement chronobiological principles into medicine. The authors bring the concept of circadian phase maps taxonomy, which represents a mechanism that determines circadian clocks of an individual and their ability to be synchronized. Firstly, standardized and reliable biomarkers of these phase maps must be set. This knowledge will allow to prescribe specific exposures of zeitgebers. New therapies could target the circadian system and personalize the therapeutic schedules for any individual.

## Figures and Tables

**Figure 1 ijms-21-03005-f001:**
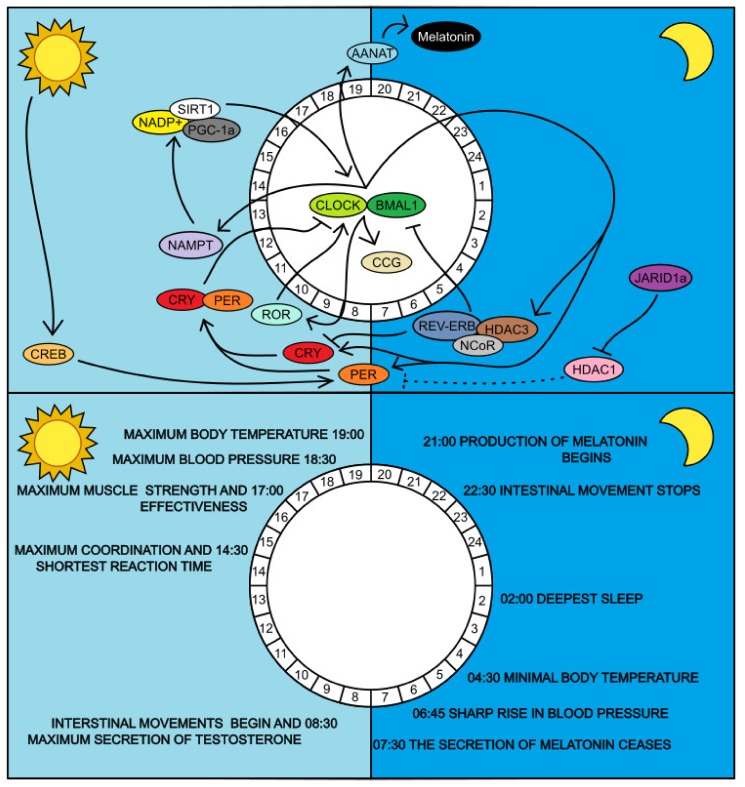
Molecular loops of circadian rhythm. Upper diagram shows interconnection of CCGs in relation to time period. CLOCK-BMAL1 complex activates the expression of PER, CRY, REV-ERB, ROR and other CCGs containing an E-box sequence in their promoter region. Subsequently, PER and CRY form a core PER-CRY complex that inhibits the CLOCK-BMAL1 complex. The expressed REV-ERB complexed with NCOR and HDAC3 proteins inhibit BMAL1 protein expression. The ROR protein competes with this complex for RRE sequence in the BMAL1 promoter and activates its expression. To avoid the permanent suppression of the major genes CLOCK, BML1 and their products, there are many mechanisms that promote their reactivation. JARID1a (KDM1A) blocks HDAC1 that maintains the PER and CRY inhibitory status. Perhaps the most important player in the field of circadian loop control is SIRT1. This, together with PGC-1α activates the expression of BMAL1 and in addition disrupts the PER-CRY complex, which then loses its inhibitory function. This regulates itself because its activity is directly dependent on NAD^+^ concentration. This concentration is maintained by NAMPT and its expression is activated by the CLOCK-BMAL1 complex [3]. Lower diagram shows physical outputs of circadian rhythm that is controlled by epigenetic molecular loops [7].

**Table 1 ijms-21-03005-t001:** Summary of possible effects of CCGs on health

Name of CCG	Manifestation of the Disease
CLOCK	Hyperphagia, hyperlipidemia, hyperinsulinemic hyperglycemia, sleep disorders [26]
BMAL1	Induction of adipogenesis in adipocytes of adipose tissue, hypotension [27,28]
PPARγ	Reduction of circadian blood pressure oscillation [29]
CRY	Hypertension, aldosterone overproduction [30]
HDAC3	Disruption in lipid homeostasis of the liver and severe steatosis [31]
PER	Circadian period change, Advanced Sleep-Wake Phase Disorder (ASWPD), Delayed Sleep-Wake Phase Disorder (DSWPD) [23,32,33]
SIRT1	Regulation of adipocyte differentiation, gluconeogenesis and insulin secretion, changes in fat burning during sleep [34,35,36]

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
