# Peer review of "Epigenetic Regulation of Circadian Rhythm and Its Possible Role in Diabetes Mellitus"

_ijms, 2020, doi:10.3390/ijms21083005_

Round 1

Reviewer 1 Report

Hudec et al well summarized the topics on clock genes. However, the reference are relatively old. There is a little recent information.

The link between disruption of circadian rhythm and onset of diabetes has been described. However, the effects of epigenetic changes on the link are not well understood yet. If the authors feel some difficulties, change of the title (or scope) could be a one of choices.

Author Response

Review Article: Epigenetic regulation of circadian rhythm and its possible role in diabetes Mellitus

To Whom It May Concern,

First, I would like to thank you very much for review comments that were very useful and helped me significantly to improve the manuscript. I have accepted all of them.

I am sending you my answer to your suggestions.

You asked me to add more recent publications. So, I followed your recommendation. I have added 20 recent publications.

Article was reviewed by a native speaker. 

Thank you again for your recognition of our research.

Best regards,

Michael Hudec

Reviewer 2 Report

GENERAL: The title announces DM and circadian rhythm relationship to be explored, but it seems they try to address the full spectrum of metabolic syndrome. In present form, the part of the manuscript presenting the pathophysiological connections is insufficient. I suggest the authors look up to papers like (http://www.nature.com/articles/s41574-018-0122-1 , https://jme.bioscientifica.com/view/journals/jme/52/1/R1.xml , https://joe.bioscientifica.com/view/journals/joe/230/1/R1.xml ). 

At times, langugage is an issue (marked in attached pdf).

The image and the table are really good.

SPECIFIC:

pg5 148-153 claim needs references. Also primary preference leads to problems and 'night owls' have dysregulation despite following their preferred rhythm (see articles recommended)

Format under the image is not right (text continues with a line and then the description follows). 

Author Response

Review Article: Epigenetic regulation of circadian rhythm and its possible role in diabetes Mellitus

To Whom It May Concern,

First, I would like to thank you very much for review comments that were very useful and helped me significantly to improve the manuscript. I have accepted all of them.

I am sending you my answer to your suggestions.

You suggested to look up into 3 articles. So, I took them into consideration and I tried to make sufficient pathophysiological connections.

Then, I changed pg5 148-153 to: It is important to note that the natural preference for a certain circadian period does not necessarily lead to health problems. The problem arises when particular individuals are forced by their surroundings (the working environment) to disregard a circadian period that is natural to them. But natural preference of eveningness (Delayed Sleep-Wake Phase Disorder) leads to dysregulation despite following preferred rhythm. It was found that each hour delay in mid-sleep time on free days (a metric of chronotype), was associated with significantly higher HbA1c. Later chronotype and larger dinner are associated with poorer glycemic control [60]. Further, it was observed that delayed sleep phase leads to a higher prevalence of type 2 diabetes, higher risk of arterial hypertension and lower fasting serum levels of total and LDL cholesterol [61].

I am sorry for the wrong format under the image. The format was optimal in our computer. So, it may be due to different versions of Microsoft Word or something similar. I tried to fix it by changing text flow.

Article was reviewed by a native speaker.

Thank you again for your recognition of our research.

Best regards,

Michael Hudec

Reviewer 3 Report

This paper addresses a very timely and important topic. In general, the paper needs to be reviewed for English language accuracy. Abstract needs to be rewritten as described below. There is key missing information on epigenetic regulation of physiological processes that occur with circadian rhythmicity. Many of the references are old and outdated. The authors missed several recent studies on epigenetic regulation of clock genes and associations with type 2 diabetes and cardiometabolic disease. Much of this research is published in the last decade. The authors should re-evaluate the available evidence so that the review accomplishes its goal.

ABSTRACT

The abstract is poorly constructed and does not provide the reader with any information on 1) the public health relevance of the topic…why is it important to understand epigenetic regulation of circadian rhythms in the context of type 2 diabetes prevention?, 2) what did the authors actually find in their review of the literature?, and 3) what are the necessary lines of research to advance this research area?

The first sentence of the abstract is simplistic and could easily be deleted.

The last sentence should be the first sentence.

Line 18 “Because it is regulated by a certain time period”….this is not a full sentence, this is a phrase. English grammar needs substantial revision for this article.

Abstract needs to be rewritten. It has too many grammatical errors, but more importantly does not convey any important information.

FIGURE 1

Where are the references for these processes? The footnotes can have the references where the authors obtained the times at which these physiological processes occur.

Please correct spelling of testosterone and muscle strength in the figure. There are typos.

INTRODUCTION

Line 26: As an individual develops and grows, their cells divide and differentiate

Lines 41-42: clarify what is meant by performing under “optimal conditions”. You give an example for impaired glucose tolerance later but it is unclear why that happens and why glucose tolerance diminishes throughout the day and why that could possibly influence the timing of meals.

Line 52: should be “can function optimally”

After reading the introduction, I still have no information from authors about why this is an important topic for public health and prevention or management of diabetes. It is not clear what type of a review this is: narrative, systematic (not likely), perspective, etc..?

What is the objective of this review and what type of evidence is reviewed here? Is your goal to highlight research gaps in this topic area?

The introduction reads like any paragraph in the paper and does not do a good job of describing why these physiological processes are important to decipher.

Your introduction should summarize what topics you will discuss in sections below and why you are focusing on these particular topics.

Epigenetic Control of Circadian Rhythms

This section is missing key information. While it does indicate that CLOCK, BMAL1, and SIRT1 are important for regulating circadian rhythms and describes transcription/translation processes, it does not clearly state their function. The reader would also be interested in learning how DNA methylation and other epigenetic changes influence physiological processes that occur with circadian rhythmicity (e.g. blood glucose regulation).

How do these epigenetic pathways influence sleep/wake cycles?

What factors influence epigenetic regulation of these clock genes?

There is emerging evidence that lifestyle behaviors influence epigenetic regulation of these genes and this should be summarized here. What is the influence of lifestyle behaviors on epigenetic regulation of these genes?

Aging of Epigenetic Mechanisms

Again here, there is some much potential to include cutting-edge research. We know that aging can lead to less robust circadian rhythms but the epidemiological data on this is not described. How do DNA methylation patterns influence circadian rhythms? Is it a bi-directional association? What is the connection to physiological processes related to chronic disease?

The review is too focused on reactions and mechanistic pathways without making the actual link to circadian rhythms and associated chronic disease risk.

Interindividual variability of circadian preference

What are the health consequences of epigenetic regulation of the genes involved in circadian rhythms? Do we know what hypermethylation or hypomethylation in PER for example does to circadian rhythms, sleep patterns, glycemic regulation, etc..? Same questions for other genes involved in regulating circadian rhythms. These are the types of data the reader is expecting to see summarized.

HOW DOES HYPER OR HYPO METHYLATION OF CCGs in TABLE 1 INFLUNECE THE DESCRIBED HEATH CONDITIONS (e.g. increase risk for hypertension or type 2 diabetes)?

Metabolic syndrome, T2DM and circadian rhythm disorders

 Line 214: Metabolic syndrome is not a new disease. Please delete this.

Line 224L Diabetes is not an outbreak. Please use appropriate scientific terminology.

Correlation is different from association. There are multiple studies and reviews/meta-analyses that demonstrate associations between poor sleep and risk for type 2 diabetes and metabolic syndrome. Your references are outdated. For a review published in 2020 it is strange that much of your reference list is from the 1990s and early 2000s. There has been so much advancement in the field in the last decade. Your review should reflect that.

Circadian rhythm of blood pressure

Line 269-270: Several aspects of sleep are convincingly linked to higher risk of hypertension. Therefore the statement: “The worse is the quality of sleep, probably, the higher is the risk of hypertension” is not really accurate.

Again, in these last 3 sections (lines 213-272), there is a discussion of what mutations in some CCGs would lead to. This is NOT epigenetic regulation. Where is all the recent data on epigenetic regulation of CCGs and their role in type 2 diabetes and hypertension etiology?

Conclusion

Conclusion seems unrelated to the topic of the review. Why is it discussing DNA methylation of CCGs in relation to tumor development when the topic of the review is epigenetic regulation of CCGs in the context of type 2 diabetes?

Where is the summary of the evidence on epigenetic pathways of CCGs and glycemic regulation and/or type 2 diabetes etiology.

The review does not accomplish what appears to be its goal. The authors have missed several landmark studies including recently published population-based studies.

Author Response

Review Article: Epigenetic regulation of circadian rhythm and its possible role in diabetes Mellitus

To Whom It May Concern,

First, I would like to thank you very much for review comments that were very useful and helped me significantly to improve the manuscript. I have accepted all of them.

I am sending you my answer to your suggestions.

You asked me to make major changes. I tried to solve the maximum of requests:

ABSTRACT

The Abstract was completely rewritten. First sentence was deleted. Last sentence was changed and moved to the beginning.

FIGURE 1

I added the references to clarify the information. I corrected the spelling of testosterone and muscle strength.

INTRODUCTION

Line 26: As an individual develops and grows, their cells divide and differentiate. This sentence was changed.

Lines 41-42: I added some examples to clarify the term of optimal conditions. First example, the levels of cortisol, important hormone in endocrine system, differ between chronotypes and have influence on sleep. Later is explained that poor sleep quality is associated with T2DM and other diseases. Second example, the levels of melatonin have effect on health condition. So, it demonstrates that physiological processes must take place under optimal conditions.

Line 52: should be “can function optimally”. This sentence was changed.

I tried to add examples about important connections between external factors, circadian clocks (central and peripheral), pathophysiological connections and aging. I tried to make a summary of topics without changing the meaning of the whole text.

Epigenetic Control of Circadian Rhythms

I added a section where I mentioned examples of factors (including lifestyle) that influence DNA methylation of circadian genes. I added also a new evidence about circadian disruption condition.

Aging of Epigenetic Mechanisms

I added a new section at the end of this topic. I summarized the effect of long term lifestyle behavior on aging. I admit there is much more potential in this topic. I have wanted just to mention that aging has an important role in epigenetic changes. But, this topic could be an independent review article.

Interindividual variability of circadian preference

Here, I added consequences of DNA methylation changes that involve circadian rhythms. Specifically, I added comments regarding CLOCK, BMAL and PER methylation.

Metabolic syndrome, T2DM and circadian rhythm disorders and Circadian rhythm of blood pressure

I tried to make some corrections to improve these topics as much as I could.

Line 214: Metabolic syndrome is not a new disease. Please delete this. This sentence was changed.

Line 224L Diabetes is not an outbreak. Please use appropriate scientific terminology. The terminology was changed.

Conclusion

Here, I added a summary about new advances in the field of T2DM, epigenetics and circadian rhythm with reference to pathophysiological conditions.

Article was reviewed by a native speaker.

Thank you again for your recognition of our research.

Best regards,

Michael Hudec

Round 2

Reviewer 1 Report

No further action is required.

Author Response

To Whom It May Concern,

I would like to thank you for your review comments.

Best regards,

Michael Hudec

Reviewer 2 Report

I thank the authors for incorporating the changes.

Page 8 - row 214 - I do not think we can claim metaoblic syndrome is a new disease

  • same row - 'signs' instead of 'singns' - the latter spelling is in the manuscript

The paper has an ambitious title, I am not entirely convinced it has addressed the issue fully.

For example, regarding blood pressure - How does sleep disturbance with sleep apnea relate to hypertension, since quality of sleep was taken as a factor (Pg 9 row 268-272). I leave to the editor to decide on the appropriate scope and size of the manuscript, but the title is very general, and the content does not entirely cover the available (and known) pathophysiological routes. Indeed, it is a complex topic.

Author Response

To Whom It May Concern,

First, I would like to thank you for your review comments, that were very useful and helped me significantly to improve the manuscript. I have accepted all of them.

I am sending you my answer to your suggestions.

Page 8 - row 214 - I do not think we can claim metabolic syndrome is a new disease, same row - 'signs' instead of 'singns' - the latter spelling is in the manuscript. The errors were revised in previous version.

We decided to change the title to better cover content to: Epigenetic regulation of circadian rhythm and its possible impact on health

Thank you again for your recognition of our research.

Best regards,

Michael Hudec

Reviewer 3 Report

Thank you for addressing the comments. The paper still needs review for English language grammar and for accuracy of content. It would be very helpful to list each comment with the response below it and to include the line numbers to indicate where the change is actually reflected.

ABSTRACT

T2DM is the abbreviation for type 2 diabetes mellitus NOT diabetes mellitus type 2

Although the abstract has been rewritten, it does not convey helpful information. Defining genetic and epigenetic mechanisms simplistically is not helpful. It would be more informative to actually summarize what you found in your literature search on the state of the evidence. 

There are some conceptual errors as well in the abstract:

1) chronotype does not determine our preference of daily rhythm. chronotype is the time of day preference for daily activities that we can think of as the phenotypic expression of our innate circadian rhythm. It is typically self-reported and therefore subjective.

2) "shifting daily rhythm preference to eveningness" does not really make sense. Chronotype is the behavioral manifestation of our innate circadian rhythm and given that an evening chronotype is generally detrimental I do not think that anyone is "shifting" to that. 

There are nuances that are not well captured by the style of writing

Your last sentence should be change to the actual conclusion. Do not say what the conclusion is going to do, just state what it actually is. What is the take home message of your review?

Introduction:

lines 95-98: doesn't the circadian system control endocrine function and metabolic processes? Your sentence indicates that the reverse is the case.

line 101: chronotype comes out of the blue with no prior introduction. Also you are mixing the concept of chronotype with circadian rhythm

there is no logical order to the flow of ideas in your introduction. You jump from one thought to another. Tell a story.

Please refrain from using conversational words such as "of course" and "obviously" throughout the manuscript.

line 120 chronotype IS a preference....please make sure that you understand and convey to the reader that chronotype, sleep patterns, and circadian rhythms are related but distinct concepts. The meaning is being lost as written.

Aging comes out of the blue. Again, tie it in appropriately to what you are trying to say. Why do circadian rhythms become less robust with aging and how does that influence your outcome of type 2 diabetes? Is it a bi-directional association?

Epigenetic control of circadian rhythms

General comment: I am not seeing in this section how lifestyle behaviors which entrain circadian rhythm (especially food signals but also sleep cycles and rest-activity patterns) lead to epigenetic changes in these genes, which can then influence T2D risk. This is a very important point that needs to addressed.

Aging of Epigenetic Mechanisms

Please change the title of this section....epigenetic mechanisms do not age

Change to: Epigenetic Changes in Clock Genes During Aging

or Circadian Rhythms and Aging

The beginning of this paragraph should define aging and then discuss how it manifests in epigenetic pathways that influence circadian rhythm.

There needs to be some acknowledgment that the association between aging and circadian rhythms is bi-directional. On one hand, less robust circadian rhythms lead to accelerated biological aging. On the other hand, aging leads to less robust circadian rhythms. Indicate how changes in sleep across the life cycle can influence that.

Interindividual variability of circadian preference

Please make sure that you clarify the difference between subjective chronotype and circadian rhythm disorders. When you assess chronotype: you will see that different populations report very different estimates of evening preference and that there is a spectrum of: definite evening, moderate evening, intermediate, moderate morning, and definite morning chronotypes. This will be different from human studies of circadian phase. It is important to make this distinction in methodology and in results and to discuss these studies separately in different paragraphs.

Please cite these more recent studies on chronotype in addition to reference 60:

Knutson KL, von Schantz M. Associations between chronotype, morbidity and mortality in the UK Biobank cohort. Chronobiology international. 2018 Aug 3.

Makarem N, Paul J, Giardina EG, Liao M, Aggarwal B. Evening chronotype is associated with poor cardiovascular health and adverse health behaviors in a diverse population of women. Chronobiology International. 2020 Mar 5:1-3.

Also discuss that chronotype may lead to cardiometabolic dysfunction due to adverse health behaviors in evening chronotypes

Metabolic syndrome, T2DM and circadian rhythm disorders

You forgot to mention that low HDL is part of MetS

line 628: there is a typo, change "develope" to "develop"

The American Heart Association in 2016 and 2017 released two statements on sleep and meal timing in relation to cardiometabolic health. These are important to look at to find key references that are related to your topic.

See the statements below:

St-Onge MP, Grandner MA, Brown D, Conroy MB, Jean-Louis G, Coons M, Bhatt DL. Sleep duration and quality: impact on lifestyle behaviors and cardiometabolic health: a scientific statement from the American Heart Association. Circulation. 2016 Nov 1;134(18):e367-86.

St-Onge MP, Ard J, Baskin ML, Chiuve SE, Johnson HM, Kris-Etherton P, Varady K. Meal timing and frequency: implications for cardiovascular disease prevention: a scientific statement from the American Heart Association. Circulation. 2017 Feb 28;135(9):e96-121.

It is relevant to mention studies of how timing of food intake and poor sleep through circadian misalignment lead to type 2 diabetes and then link this to the epigenetic pathways that you discuss to paint a full picture. That would make this section much more coherent.

The section on circadian rhythm and blood pressure is missing key information. Cite the following reviews of epidemiological data which show that sufficient sleep, fixed sleep schedules, and absence of social jet lag could lower the risk of circadian misalignment and therefore lower the risk for hypertension, which also represents a risk factor for MetS and T2D. These are the references:

Makarem N, Shechter A, Carnethon MR, Mullington JM, Hall MH, Abdalla M. Sleep duration and blood pressure: Recent advances and future directions. Current hypertension reports. 2019 May 1;21(5):33.

Makarem N, Zuraikat FM, Aggarwal B, Jelic S, St-Onge MP. Variability in Sleep Patterns: an Emerging Risk Factor for Hypertension. Current Hypertension Reports. 2020 Feb;22(2):1-0.

Conclusion

Please again more comprehensively summarize lifestyle behaviors that lead to circadian misalignment (not just sleep quality...also meal timing, sleep duration and timing, variability in lifestyle behaviors, etc..) and epigenetic changes in circadian clock genes and indicate how improvement in lifestyle approaches may inform precision behavioral approaches for prevention. 

Author Response

To Whom It May Concern,

First, I would like to thank you for your review comments that were very useful and helped me significantly to improve the manuscript. I have accepted all of them.

I am sending you my answer to your suggestions.

You asked me to make major changes. I tried to solve the maximum of requests:

ABSTRACT

T2DM is the abbreviation for type 2 diabetes mellitus NOT diabetes mellitus type 2. The abbreviation was revised.

-Lines 32-34: The text was rewritten.

Introduction:

lines 95-98: doesn't the circadian system control endocrine function and metabolic processes? Your sentence indicates that the reverse is the case.

-Lines 55-58: The text was revised.

line 101: chronotype comes out of the blue with no prior introduction. Also you are mixing the concept of chronotype with circadian rhythm

-Lines 59-63: The text was revised.

Aging comes out of the blue. Again, tie it in appropriately to what you are trying to say. Why do circadian rhythms become less robust with aging and how does that influence your outcome of type 2 diabetes? Is it a bi-directional association?

-Lines 78-80 and 193-200: The text was revised.

Epigenetic control of circadian rhythms

General comment: I am not seeing in this section how lifestyle behaviors which entrain circadian rhythm (especially food signals but also sleep cycles and rest-activity patterns) lead to epigenetic changes in these genes, which can then influence T2D risk. This is a very important point that needs to addressed.

-Lines 285-288 and 294-296: The text was revised in different section.

Aging of Epigenetic Mechanisms

Please change the title of this section....epigenetic mechanisms do not age

-The title of this section was changed to: Circadian Rhythms and Aging

There needs to be some acknowledgment that the association between aging and circadian rhythms is bi-directional. On one hand, less robust circadian rhythms lead to accelerated biological aging. On the other hand, aging leads to less robust circadian rhythms. Indicate how changes in sleep across the life cycle can influence that.

-Lines: 193-200: The text was revised.

Interindividual variability of circadian preference

Knutson KL, von Schantz M. Associations between chronotype, morbidity and mortality in the UK Biobank cohort. Chronobiology international. 2018 Aug 3.

Makarem N, Paul J, Giardina EG, Liao M, Aggarwal B. Evening chronotype is associated with poor cardiovascular health and adverse health behaviors in a diverse population of women. Chronobiology International. 2020 Mar 5:1-3.

-Thank you for suggested references. I took them into consideration.

Metabolic syndrome, T2DM and circadian rhythm disorders

St-Onge MP, Grandner MA, Brown D, Conroy MB, Jean-Louis G, Coons M, Bhatt DL. Sleep duration and quality: impact on lifestyle behaviors and cardiometabolic health: a scientific statement from the American Heart Association. Circulation. 2016 Nov 1;134(18):e367-86.

St-Onge MP, Ard J, Baskin ML, Chiuve SE, Johnson HM, Kris-Etherton P, Varady K. Meal timing and frequency: implications for cardiovascular disease prevention: a scientific statement from the American Heart Association. Circulation. 2017 Feb 28;135(9):e96-121.

-Thank you for suggested statements, I took them into consideration.

It is relevant to mention studies of how timing of food intake and poor sleep through circadian misalignment lead to type 2 diabetes and then link this to the epigenetic pathways that you discuss to paint a full picture. That would make this section much more coherent.

The section on circadian rhythm and blood pressure is missing key information. Cite the following reviews of epidemiological data which show that sufficient sleep, fixed sleep schedules, and absence of social jet lag could lower the risk of circadian misalignment and therefore lower the risk for hypertension, which also represents a risk factor for MetS and T2D. These are the references:

Makarem N, Shechter A, Carnethon MR, Mullington JM, Hall MH, Abdalla M. Sleep duration and blood pressure: Recent advances and future directions. Current hypertension reports. 2019 May 1;21(5):33.

Makarem N, Zuraikat FM, Aggarwal B, Jelic S, St-Onge MP. Variability in Sleep Patterns: an Emerging Risk Factor for Hypertension. Current Hypertension Reports. 2020 Feb;22(2):1-0.

-Thank you for suggested references. I took them into consideration.

-Lines 345-346: The text was revised.

Conclusion

Please again more comprehensively summarize lifestyle behaviors that lead to circadian misalignment (not just sleep quality...also meal timing, sleep duration and timing, variability in lifestyle behaviors, etc..) and epigenetic changes in circadian clock genes and indicate how improvement in lifestyle approaches may inform precision behavioral approaches for prevention.

Lines 352-358: The text was rewritten

Lines 378-380, 394-395 and 397-399: The text was revised.

We decided to change the title to better cover the content to: Epigenetic regulation of circadian rhythm and its possible impact on health

Thank you again for your recognition of our research.

Best regards,

Michael Hudec

Round 3

Reviewer 2 Report

I believe the paper shows good basic concepts on circadian biology and has nice images that clearly depict the stated. That is a strength.

I believe the paper has been re-titled with a broader scope. This is rather an ambitious undertaking and only certain aspects of pathophysiology with clinical implications were presented. 

I do not see this as a piece that would advance the filed or get much appreciation from scientific community. There are topical review papers that cover the topics more broadly, yet are more specific (some appear as references in this manuscript).

Author Response

To Whom It May Concern,

First, I would like to thank you for your review comments. I have accepted all of them.

I am sending you my answer to your suggestions:

I believe the paper shows good basic concepts on circadian biology and has nice images that clearly depict the stated. That is a strength.

I believe the paper has been re-titled with a broader scope. This is rather an ambitious undertaking and only certain aspects of pathophysiology with clinical implications were presented.

I do not see this as a piece that would advance the filed or get much appreciation from scientific community. There are topical review papers that cover the topics more broadly, yet are more specific (some appear as references in this manuscript).

After revision, we decided to change back the title to: Epigenetic regulation of circadian rhythm and its possible role in diabetes mellitus.

ijms-715956-peer-review-r2, between sections 1. Introduction, line 129 and 2. Epigenetic control of circadian rhythms: line 120: We added new sections: Genetics of type 2 diabetes mellitus and Epigenetics of type 2 diabetes mellitus. These new sections are designed to better fill the content of this review.

ijms-715956-peer-review-r2, section 6. Conclusion and possible treatments, between lines 497 and 498: We added a new paragraph summarizing the environmental impact on T2DM, again, to better fill the content of this review.  

Thank you again for your recognition of our research.

Best regards,

Michael Hudec

Reviewer 3 Report

Thank you for addressing comments. 

Introduction:

lines 49-50: what is meant by "under optimal conditions", do you mean "with circadian rhythmicity"? If so, explicitly state this because that is more accurate. My comment about chronotype was also not addressed...where is the definition of chronotype. When this concept is first introduced it should be followed by a definition (time of day preference for daily activities) 

Conclusion:

Conclusion is still incoherent, consecutive sentences are unrelated to each other. For instance you talk about chrono-nutrition in lines 365-366...what are you suggesting? That timing of meals be aligned with chronotype or innate circadian rhythm? Then, you talk about Facer-Childs...unclear what this sentence is about....what is the non-pharmacological intervention that you are referring to?. Then, you shift to chrono-medicine without indicating what you mean....are you indicating that targeting timing of prescription medication would result in more optimal disease outcomes? These ideas are all not well developed or connected as written.

First sentence of conclusion is incorrect. Lifestyle behaviors may influence epigenetic regulation of circadian rhythms....so why do you suggest that these are difficult to modify? Why do "tumors" randomly come up in the following sentence. 

The biggest challenge is the logical flow of your conclusion.

Author Response

To Whom It May Concern,

First, I would like to thank you for your review comments. I have accepted all of them.

I am sending you my answer to your suggestions:

Introduction:

lines 49-50: what is meant by "under optimal conditions", do you mean "with circadian rhythmicity"? If so, explicitly state this because that is more accurate.

ijms-715956-peer-review-r2, between sections 1. Introduction, line 83: We changed the "under optimal conditions" to "with circadian rhythmicity".

My comment about chronotype was also not addressed...where is the definition of chronotype. When this concept is first introduced it should be followed by a definition (time of day preference for daily activities)

ijms-715956-peer-review-r2, between sections 1. Introduction, line 83: We added the definition of chronotype "This is expressed as Chronotype, the time of day preference for daily activities".

Conclusion:

Conclusion is still incoherent, consecutive sentences are unrelated to each other. For instance you talk about chrono-nutrition in lines 365-366...what are you suggesting? That timing of meals be aligned with chronotype or innate circadian rhythm?

ijms-715956-peer-review-r2, section 6. Conclusion and possible treatments, lines 492 and 493: We changed the "The patient's circadian period can be optimized by setting appropriate eating and sleeping times" to "The eating and sleeping times should be aligned with patient's chronotype".

Then, you talk about Facer-Childs...unclear what this sentence is about....what is the non-pharmacological intervention that you are referring to?

ijms-715956-peer-review-r2, section 6. Conclusion and possible treatments, line 513: We added "Facer-Childs and colleagues show the ability of a simple non-pharmacological intervention reducing negative elements of mental health and disturbances of sleep by a phase advance of around 2 h. There were significant reductions in subjective ratings of depression and stress. In addition, elements of cognitive (reaction time) and physical (grip strength) performance were significantly improved during ‘non optimal’ times"

Then, you shift to chrono-medicine without indicating what you mean....are you indicating that targeting timing of prescription medication would result in more optimal disease outcomes? These ideas are all not well developed or connected as written.

ijms-715956-peer-review-r2, section 6. Conclusion and possible treatments, line 515: We added " Roenneberg and Merrow offer a plan how to implement chronobiological principles into medicine. The authors bring the concept of circadian phase maps taxonomy, which represents a mechanism that determines circadian clocks of an individual and their ability to be synchronized. Firstly, standardized and reliable biomarkers of these phase maps must be set. This knowledge will allow to prescribe specific exposures of zeitgebers. New therapies could target the circadian system and personalize the therapeutic schedules for any individual".

These two paragraphs (from line 513 and 515) are intended to describe possible outlooks for future treatments.

First sentence of conclusion is incorrect. Lifestyle behaviours may influence epigenetic regulation of circadian rhythms....so why do you suggest that these are difficult to modify? Why do "tumors" randomly come up in the following sentence.

ijms-715956-peer-review-r2, section 6. Conclusion and possible treatments, lines 443-445: We changed the sentence from "The epigenetic processes regulating circadian rhythms are difficult to modify. A problem that will remain unresolved for a long time is that these approaches are non-specific" to "The epigenetic processes regulating circadian rhythms are difficult to modify by using drugs. A problem that will remain unresolved for a long time is that approaches using DNA methylation inhibitors and histone deacetylase inhibitors are non-specific".

Here, we would like to mention that simple drug treatment at the epigenetic level is "not very suitable for the treatment of circadian rhythms dysregulation" because "these approaches are directed to the treatment of tumors".

After revision, we decided to change back the title to: Epigenetic regulation of circadian rhythm and its possible role in diabetes mellitus.

ijms-715956-peer-review-r2, between sections 1. Introduction, line 129 and 2. Epigenetic control of circadian rhythms: line 120: We added new sections: Genetics of type 2 diabetes mellitus and Epigenetics of type 2 diabetes mellitus. these new sections are designed to better fill the content of this review.

ijms-715956-peer-review-r2, section 6. Conclusion and possible treatments, between lines 497 and 498: We added a new paragraph summarizing the environmental impact on T2DM, again, to better fill the content of this review. 

Thank you again for your recognition of our research.

Best regards,

Michael Hudec

Round 4

Reviewer 3 Report

Thank you for addressing feedback.